# FakeAVCeleb: A Novel Audio-Video Multimodal Deepfake Dataset

**Hasam Khalid**[1], **Shahroz Tariq**[1], **Minha Kim**[1], **Simon S. Woo**[*,1,2,3]

[1] College of Computing and Informatics
[2] Department of Applied Data Science
[3] Department of Artificial Intelligence
Sungkyunkwan University, South Korea
{hasam.khalid, shahroz, kimminha, swoo}@g.skku.edu

## Abstract

While the significant advancements have made in the generation of deepfakes using deep learning technologies, its misuse is a well-known issue now. Deepfakes can cause severe security and privacy issues as they can be used to impersonate a person's identity in a video by replacing his/her face with another person's face. Recently, a new problem of generating synthesized human voice of a person is emerging, where AI-based deep learning models can synthesize any person's voice requiring just a few seconds of audio. With the emerging threat of impersonation attacks using deepfake audios and videos, a new generation of deepfake detectors is needed to focus on both video and audio collectively. To develop a competent deepfake detector, a large amount of high-quality data is typically required to capture real-world (or practical) scenarios. Existing deepfake datasets either contain deepfake videos or audios, which are racially biased as well. As a result, it is critical to develop a high-quality video and audio deepfake dataset that can be used to detect both audio and video deepfakes simultaneously. To fill this gap, we propose a novel Audio-Video Deepfake dataset, FakeAVCeleb, which contains not only deepfake videos but also respective synthesized lip-synced fake audios. We generate this dataset using the most popular deepfake generation methods. We selected real YouTube videos of celebrities with four ethnic backgrounds to develop a more realistic multimodal dataset that addresses racial bias, and further help develop multimodal deepfake detectors. We performed several experiments using state-of-the-art detection methods to evaluate our deepfake dataset and demonstrate the challenges and usefulness of our multimodal Audio-Video deepfake dataset.

## 1 Introduction

With the advent of new AI technologies, particularly deep neural networks (DNNs), a rise in forged or manipulated images, videos, and audios has been skyrocketed. Even though forging and manipulating images or videos has been performed in the past [1], generating highly realistic fake human face images [2] or videos [3], as well as cloning human voices [4] have become much easier and faster than before. Recently, generating deepfakes using a DNN based technique to replace a person's face with another person, has significantly increased. The most common deep learning-based generation methods make use of Autoencoders (AEs) [5], Variational Autoencoders (VAEs) [5], and Generative Adversarial Networks (GANs) [2]. These methods are used to combine or superimpose a source human face image onto a target image. In addition, the recent advancements in deep learning-based

---

[*]Corresponding author

35th Conference on Neural Information Processing Systems (NeurIPS 2021) Track on Datasets and Benchmarks.

deepfake generation methods have created voice-cloned human voices in real-time [4, 6], where human voice cloning is a network-based speech synthesis method to generate high-quality speech of target speakers [4]. One famous example of deepfakes is of former U.S. Presidents Barack Obama, Donald Trump, and George W. Bush, generated as a part of research [7]. The individuals in the videos can be seen speaking with very accurate lip-sync. As a result, deepfakes have become a challenging well-known technical, social, and ethical issue now. Numerous discussions are being held on state news channels and social media related to the potential harms of deepfakes [8]. A recent article was published on Forbes [9], which discussed a TV commercial ad by ESPN [10] that became a center of discussion over social media, where the ad showed footage from 1998 of an ESPN analyst making shockingly accurate predictions about the year 2020. It later turned out that the clip was fake and generated using cutting-edge AI technology [11]. Moreover, Tariq et al. [12] demonstrate how deepfakes impersonation attacks can severely impact face recognition technologies.

Therefore, many ethical, security, and privacy concerns arise due to the ease of generating deepfakes. Keeping in mind the misuse of deepfakes and the potential harm they can cause, the need for deepfake detection methods is inevitable. Many researchers have dived into this domain and proposed a variety of different deepfake detection methods [12–31]. First, high quality deepfake datasets are required to create an efficient and usable deepfake detection method. Researchers have also proposed different deepfake datasets generated using the latest deepfake generation methods [32–37] to aid other fellow scholars in developing deepfake detection methods. Most of the recent deepfake detection methods [16, 20, 21, 24, 28, 28–30, 38, 39] make use of these publicly available datasets. However, these deepfake datasets only focus on generating realistic deepfake videos but do not consider generating fake audio for them. While some neural network-based synthesized audio datasets exist [40–44], they do not contain the respective lip-synced or deepfake videos. This limitation of deepfake datasets also hinders the development of multimodal deepfake detection methods, which can detect deepfakes that are possibly any of audio or video combinations. And the current available datasets are limited to be used for detecting only one type of deepfakes.

As per our knowledge, there exists only one deepfake dataset, Deepfake Detection Challenge (DFDC) [45], which contains a mix of deepfake video and synthesized cloned audio, or both. However, the dataset is not labeled with respect to audio and video. It is not trivial to determine if the audio was fake or the video due to the lack of labels. Therefore, we propose a novel Audio-Video Multimodal Deepfake Detection dataset (FakeAVCeleb), which contains not only deepfake videos but also respective synthesized cloned deepfake audios. Table 1 summarizes a quantitative comparison of FakeAVCeleb with other publicly available deepfake datasets. FakeAVCeleb consists of videos of celebrities with different ethnic backgrounds belonging to diverse age groups with equal proportions of each gender. To evaluate and analyze the usefulness of our dataset, we performed the comprehensive experimentation using 11 different unimodal, ensemble-based and multimodal baseline deepfake methods [13–15, 36, 46–51]. In addition, we present the detection results with other popular deepfake datasets. The main contributions of our work are summarized as follows:

- We present a novel Audio-Video Multimodal Deepfake Detection dataset, FakeAVCeleb, which contains both, video and audio deepfakes with accurate lip-sync with fine-grained labels. Such a multimodal deepfake dataset has not been developed in the past.

- Our FakeAVCeleb dataset contains three different types of Audio-Video deepfakes, generated from a carefully selected real YouTube video dataset using recently proposed popular deepfake generation methods.

- The individuals in the dataset are selected based on five major ethnic backgrounds speaking English to eliminate racial bias issues. Moreover, we performed the comprehensive baseline benchmark evaluation and demonstrated the crucial need and usefulness for a multimodal deepfake dataset.

## 2  BACKGROUND AND MOTIVATION

There are several publicly deepfake detection datasets proposed by different researchers [14, 33, 34, 37, 45, 52]. Most of these datasets are manipulated images of a person's face, i.e., swapped with another person, and these datasets contain real and respective manipulated fake videos. Nowadays, many different methods exist to generate deepfakes [2, 18]. Recently, researchers have proposed more realistic deepfake datasets with better quality and larger quantity [36, 37, 53]. However, their focus was to generate deepfake videos and not their respective fake audios. Moreover, these datasets

Table 1: Quantitative comparison of FakeAVCeleb to existing publicly available Deepfake dataset.

| Dataset | Real Videos | Fake Videos | Total Videos | Rights Cleared | Agreeing subjects | Total Subjects | Synthesis Methods | Real Audio | Deepfake Audio | Fine-grained labeling |
|---|---|---|---|---|---|---|---|---|---|---|
| UADFV [14] | 49 | 49 | 98 | No | 0 | 49 | 1 | No | No | No |
| DeepfakeTIMIT [41] | 640 | 320 | 960 | No | 0 | 32 | 2 | No | Yes | No |
| FF++ [36] | 1,000 | 4,000 | 5,000 | No | 0 | N/A | 4 | No | No | No |
| Celeb-DF [34] | 590 | 5,639 | 6,229 | No | 0 | 59 | 1 | No | No | No |
| Google DFD [36] | 363 | 3,000 | 3,363 | Yes | 28 | 28 | 5 | No | No | No |
| DeeperForensics [33] | 50,000 | 10,000 | 60,000 | Yes | 100 | 100 | 1 | No | No | No |
| DFDC [35] | 23,654 | 104,500 | 128,154 | Yes | 960 | 960 | 8 | Yes | Yes | No |
| KoDF [37] | 62,166 | 175,776 | 237,942 | Yes | 403 | 403 | 6 | Yes | No | No |
| FakeAVCeleb | 500 | 19,500 | 20,000 | No | 0 | 500 | 4 | Yes | Yes | Yes |

either contain real audio or no audio at all. In this paper, we propose a novel deepfake audio and video dataset. We generate a cloned voice of the target speaker and apply it to lip-sync with the video using facial reenactment. As per our knowledge, this is the first of its kind dataset containing deepfake videos with their respective fake audios. We believe our dataset will help researchers develop multimodal deepfake detectors that can detect both the deepfake audio and video simultaneously.

The UADFV [14] and Deepfake TIMIT [41] are some early deepfake datasets. These datasets contain fewer real videos and respective deepfake videos, and act as baseline datasets. However, their quality and quantity are low. For example, UADFV consists of only 98 videos. Meanwhile, Deepfake TIMIT contains the audio, and the video. But their audios are real, and not synthesized or fake. In our FakeAVCeleb, we propose an Audio-Video Deepfake dataset that contains not only deepfake videos but also synthesized lip-synced fake audios.

Due to the limitations of the quality and quantity of previous deepfake datasets, researchers proposed more deepfake datasets with a large number of high quality videos. FaceForensics++ (FF++) [36] and Deepfake Detection Challenge (DFDC) [35] dataset were the first large-scale datasets containing a huge amount of deepfake videos. FF++ contains 5,000 and DFDC contains 128,154 videos which were generated using multiple deepfake generation methods (FF++: 4, DFDC: 8). FF++ used a base set of 1,000 real YouTube videos and used four deepfake generations models, resulting in 5,000 deepfake videos. Later, FF++ added two more deepfake datasets, Deepfake Detection (DFD) [36] and FaceShifter [54] datasets. On the other hand, Amazon Web Services, Facebook, Microsoft, and researchers belonging to academics collaborated and released Deepfake Detection Challenge Dataset (DFDC) [35]. The videos in the DFDC dataset were captured in different environmental settings and used eight types of synthesizing methods to generate deepfake videos.

So far, most deepfake detectors [16, 17, 22, 24, 28–30, 55–57] use FF++ and DFDC datasets to train their models. However, most of these datasets lack diversity as the individuals in videos belong to specific ethnic groups. Moreover, DFDC contains videos in which participants record videos, while walking and not looking towards the camera with extreme environmental settings (i.e., dark or very bright lighting conditions), making it much harder to detect. As per our knowledge, DFDC is the only dataset containing synthesized audio with the video, but they label the entire video as fake. And they do not specify if the video is fake or the audio. Furthermore, the synthesized audios are not lip-synced with the respective videos. They even label a video fake if the voice in the video was replaced with another person's voice. Meanwhile, our FakeAVCeleb addresses these issues of environmental conditions, diversity, and respective audio-video labeling, and contains real and fake videos of people with different ethnic backgrounds, ages, and gender. We carefully selected 500 videos belonging to different ages/gender and ethnic groups from the VoxCeleb2 dataset [58], consisting of a large amount of real YouTube videos.

Recently, some new deepfake datasets have come into the light. Researchers have tried to overcome previous datasets' limitations and used new deepfake generation methods to generate deepfake videos. Celeb-DF [34] was proposed in 2020, in which researchers used 500 YouTube real videos of 59 celebrities. They applied the modified version of the popular FaceSwap method [59] to generate deepfake videos. Google also proposed a Deepfake Detection dataset (DFD) [36] containing 363 real videos and 3,000 deepfake videos, respectively. The real videos belong to 28 individuals having different ages and gender. Deepfake Videos in the Wild [53] and DeeperForensics-1.0 [33] are the most recent deepfake datasets. In particular, Deepfake Videos in the Wild dataset contains 1,869 samples of real-world deepfake videos from YouTube, and comprehensively analyzes the popularity, creators, manipulation strategies, and deepfake generation methods. The latter consists of real videos recorded by 100 paid consensual actors. They used 1,000 real videos from the FF++ dataset as target videos to apply FaceSwap. On the other hand, DeeperForensics-1.0 used a single face-swapping

method and applied augmentation on real and fake videos, producing 50,000 real and 10,000 fake videos. However, all of the aforementioned datasets contain much fewer real videos than fake ones, except for DeeperForensics-1.0, which contains more real videos than fake ones.

Moreover, there exists an issue of racial bias in several deepfake datasets. The datasets that are known to be partially biased include UADFV [14], Deepfake TIMIT [41], and KoDF [37], where the KoDF dataset contains videos of people having Korean ethnicity. Similarly, UADFV mainly contains 49 real videos from YouTube, and Deepfake TIMIT mentions that it contains only English-speaking Americans. In particular, DeeperForensics-1.0 is racially unbiased since the real videos consist of actors from 26 countries, covering four typical skin tones; white, black, yellow, and brown. On the other hand, Celeb-DF has an unbalanced number of videos for different ethnic groups and gender classes, and mainly contains Western celebrities. To the best of our knowledge, no extensive study explores the racial bias in the deepfake dataset. In our FakeAVCeleb, since we selected real videos from the VoxCeleb2 dataset, which contains 1,092,009 real videos, the number of real videos is not limited to 500. Researchers can use more real videos from the VoxCeleb2 dataset to train their models if required. Moreover, all of the datasets mentioned above only contain deepfake videos and not fake audios.

For research in human voice synthesis, a variety of new research has been conducted to simulate a human voice using neural networks. Most of these models use Tacotron [60] by Google to generate increasingly realistic, human-like human voices. Also, Google proposed the Automatic Speaker Verification Spoofing (ASVspoof) [40] challenge dataset with the goal of speaker verification and spoofed voice detection. However, all of the datasets mentioned earlier either contain deepfake videos or synthesized audios but not both. In our FakeAVCeleb, we propose a novel deepfake dataset containing deepfake videos with respective lip-synced synthesized audios. To generate fake audios, we used a transfer learning-based real-time voice cloning tool (SV2TTS) [4] that takes a few seconds of audio of a target person along with text and generates a cloned audio. Moreover, each sample in our fake audios is unique, since we clone the audio of every real video we have in our real video set. Hence, our dataset is unique and better, as it contains fake audios of multiple speakers.

## 3   Dataset Collection and Generation

**Real Dataset Collection.**   To generate our FakeAVCeleb, we gathered real videos from the Vox-Celeb2 [58] dataset, where VoxCeleb2 consists of real YouTube videos of 6,112 celebrities. It contains 1,092,009 videos in the development set and 36,237 in the test set, where each video contains interviews of celebrities and the speech audio spoken in the video. We chose 500 videos from VoxCeleb2 with an average duration of 7.8 seconds, one video for each celebrity to generate our FakeAVCeleb dataset. All the selected videos were kept with the same dimension as in the VoxCeleb2 dataset. Since the VoxCeleb2 dataset is relatively gender-biased, we tried to select videos equally based on gender, ethnicity, and age. The individuals in the real video set belong to the different ethnic groups, Caucasian, Black, South Asian, and East Asian. Each ethnic group contains 100 real videos of 100 celebrities. The male and female ratio of each ethnic group is 50%, i.e., 50 videos of men and 50 videos of women out of 100 videos.

After carefully watching and listening to each of the sampled videos, we incorporated 500 unique real videos to our FakeAVCeleb as a real baseline video set, each belonging to a single individual, with an average of 7.8 seconds duration. Since we focus on a specific and practical usage of deepfakes, each video was selected based on some specific criteria, i.e., there is a single person in the video with a clear and centered face, and he or she is not wearing any hat, glasses, mask or anything that might cover the face. Furthermore, the video quality should be good, and he or she must speak English, regardless of their ethnic background. Since we selected videos from the VoxCeleb2 dataset, which consists of real videos, more videos of the same celebrity can be selected if required to increase the number of real videos set, as we provide the original celebrity IDs used in the VoxCeleb2 dataset. Due to this reason, our FakeAVCeleb dataset is highly scalable, and we can easily generate more deepfake videos to increase the number of real and deepfake videos if required.

**Deepfake Dataset Generation.**   We used the latest deepfake and synthetic voice generation methods to generate our FakeAVCeleb dataset. We used face-swapping methods, Faceswap [59], and FSGAN [61], to generate swapped deepfake videos. To generate cloned audios, we used a transfer learning-based real-time voice cloning tool (SV2TTS) [4], (see Figure 2). After generating fake

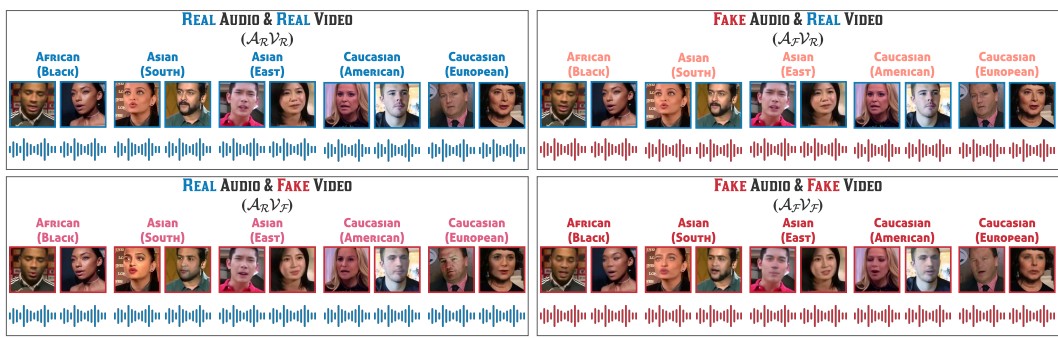

Figure 1: Samples from the Dataset. We divide the dataset into 5 ethnic groups Black, South Asian, East Asian, Caucasian (American) and Caucasian (European). There are total 4 combinations of our dataset: $\mathcal{A_R V_R}$ (500), $\mathcal{A_F V_R}$ (500), $\mathcal{A_R V_F}$ (9,000), and $\mathcal{A_F V_F}$ (10,000).

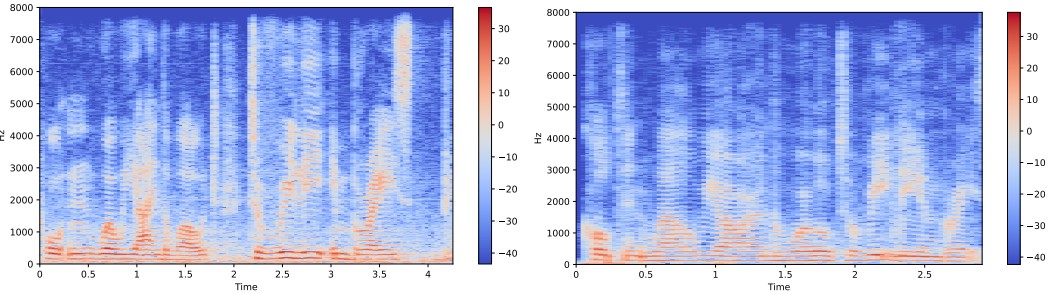

Figure 2: Sample spectrogram of real audio $\mathcal{A_R}$ (left) and fake audio $\mathcal{A_F}$ (right).

videos and audios, we apply Wav2Lip [62] on generated deepfake videos to reenact the videos based on generated fake audios. As a result, we have a fake video with a fake audio dataset that is perfectly lip-synced. This type of FakeAVCeleb dataset is minacious, as an attacker can generate fake video and fake audio and impersonate any potential target person. It is unique of its kind as all of the previous datasets either contain deepfake videos or synthesized audios. This type of deepfake dataset can be used for training a detector for both, deepfake video and deepfake audio datasets.

To generate deepfake videos, we used 500 real videos as a base set and generated around 20,000 deepfake videos using several deepfake generation methods, including face-swapping and facial reenactment methods. We also used synthetic speech generation methods to generate cloned voice samples of the people in the videos. Throughout the following sections, we will use the term *source* to refer to the source video from which we will extract the frames, and the term *target* will refer to the target video in which the face from the extracted frames will be swapped. To generate deepfake videos, we use Faceswap [59], and Faceswap GAN (FSGAN) [61] to perform face swaps, and use Wav2Lip [62] for facial reenactment based on source audio. On the other hand, the Real-Time Voice Cloning tool (RTVC) such as SV2TTS [4] is used for synthetic cloned voice generation.

Since our real video set contains people from four different ethnic backgrounds, we apply above mentioned chosen synthesis methods for each ethnicity separately (i.e., Caucasian with Caucasian or Asian with Asian). In addition, we apply synthesis for each gender separately (i.e., Men with Men and Women with Women). We applied such mappings to make a more realistic and natural fake dataset, since cross-ethnic and cross-gender face swaps result in non-natural poor deepfakes, thereby they are easily detectable. Moreover, we use a facial recognition service called Face++ [63], which measures the similarity between two faces. The similarity score helps us find the most similar source and target pairs, resulting in more realistic deepfakes. We used the Face++ API to measure the similarity of a face in a source video with faces in many other target videos. We selected the top 5 videos with the highest similarity score. After calculating the similarity score, each video was synthesized with the chosen five videos by applying synthesis methods to produce high quality realistic deepfakes. In particular, we use a total of 4 different video and audio synthesis methods. After applying all methods on each input video, the total number of fake videos comes out to be more than $20,000$.

After applying synthesis methods to generate deepfake videos, these videos are manually inspected by our researchers. Since the generated video count is high and removing bad samples manually is

Table 2: All possible combinations of real and fake video and audio datasets that we covered in our FakeAVCeleb.

| Dataset | Real Audio ($\mathcal{A_R}$) | Fake Audio ($\mathcal{A_F}$) |
|---|---|---|
| **Real Video** ($\mathcal{V_R}$) | $\mathcal{A_R V_R}$: VoxCeleb2 | $\mathcal{A_F V_R}$: SV2TTS |
| **Fake Video** ($\mathcal{V_F}$) | $\mathcal{A_R V_F}$: FSGAN, FaceSwap, Wav2Lip | $\mathcal{A_F V_F}$: FSGAN, FaceSwap, Wav2Lip, SV2TTS |

very difficult, we created a subset of the videos with respect to each fake type and then performed the inspection. While inspecting the generated videos, we filter the videos according to the following criteria: 1) The resulting fake video must be of good quality and realistic, i.e., hard to detect through the human eye, 2) The synthesized cloned audio should also be good i.e., the text given as input is synthesized properly, 3) The video and corresponding audio should be lip-synced. After the inspection, the final video count is 20,000. We found that some of the synthesis methods, FSGAN and Wav2Lip, resulted in many fake videos with excellent and realistic quality. Meanwhile, FaceSwap produced several defective videos, since it is sensitive to different lightning conditions. Also, it requires excessive time and resources to train. Some of the frames from the final real and fake videos are shown in Figure 1. We provide details of the audio and video synthesis methods used for FakeAVCeleb Section B in Appendix.

### 3.1 FakeAVCeleb Dataset Description

In this section, we discuss dataset generation using the aforementioned synthesis methods and explain the resulting types of deepfake datasets. Since we are generating cloned voices along with the fake video, we can create four possible combinations of audio-video pairs as shown in Table 2. Also, a pictorial representation of these four combinations is presented in Figure 1. The top-left images in Figure 1 are the samples from our base dataset, which contains real videos with real audio ($\mathcal{A_R V_R}$). The top-right images are the samples from real video with fake audio ($\mathcal{A_F V_R}$), which are developed using the cloned speech of the target speaker. And the bottom-left images are the samples with fake videos with real audio ($\mathcal{A_R V_F}$), representing the existing deepfakes benchmark datasets such as FF++. The bottom-right images are samples with fake video and fake audio ($\mathcal{A_F V_F}$), representing the main contribution of our work. We will explain these combinations and their respective use-cases below:

**Real-Audio and Real-Video ($\mathcal{A_R V_R}$).** We sampled 500 videos with diverse ethnicity, gender, and age. This dataset is the base set of real videos with real audios that we selected from the VoxCeleb2 dataset. Since VoxCeleb2 contains more than a million videos, more real videos from the VoxCeleb2 dataset can be utilized to train a deepfake detector for real audios and real videos (see $\mathcal{A_R V_R}$ in Table 2).

**Fake-Audio and Real-Video ($\mathcal{A_F V_R}$).** This deepfake type contains the cloned fake audio of a person along with the real video. We generate cloned fake audio using a transfer learning-based real-time voice cloning tool (SV2TTS), which takes real audio and text as an input, and outputs synthesized audio (with voice matching) of the same person, as shown in Table 2 ($\mathcal{A_F V_R}$). Please refer to the top-right block in Figure 1 for some sample results, and Figure 2 for spectrograms from real and fake audio. Since we do not have the text spoken in the video, we used IBM Watson speech-to-text service [64] that converts audio into text. This text, along with the corresponding audio, is then passed to the SV2TTS. Later, we merge the synthesized audio with the original video, resulting in the $\mathcal{A_F V_R}$ pair (see Figure 3). As a result, we were able to generate 500 fake videos. Note that our pipeline is not dependent on IBM Watson speech-to-text service; any speech-to-text service can be used as we are just extracting text from our audios. Since it is impossible to generate fake audio with the same timestamp as the original audio, this type of deepfake is not lip-synced. The possible use-case of this type of deepfake is a person performing identity fraud by impersonating a person or speaker recognition system. This dataset type can be also used to defend against anti-voice spoofing attacks, since we have real-fake audio pairs with similar text.

**Real-Audio and Fake-Video ($\mathcal{A_R V_F}$).** This type of deepfake consists of a face-swap or face-reenacted video of a person along with the real audio. To generate deepfake videos of this type, we employ three deepfake generation methods, FaceSwap and FSGAN for face-swapping, and Wav2Lip for audio-driven facial reenactment $\mathcal{A_R V_F}$ as shown in Table 2.

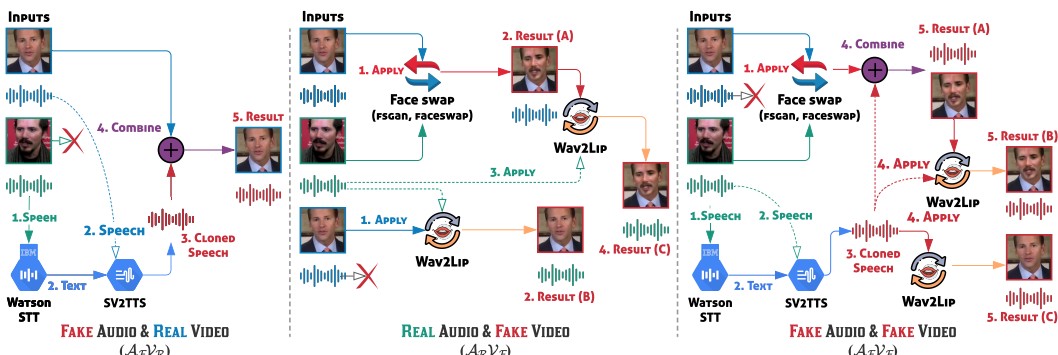

Figure 3: A step-by-step description of our FakeAVCeleb generation pipeline. The first, second, and the third method represents $\mathcal{A_F V_R}$, $\mathcal{A_R V_F}$, and $\mathcal{A_F V_F}$ generation methods, respectively, where the second ($\mathcal{A_R V_F}$) and the third method ($\mathcal{A_F V_F}$) contains lip-synched, and the first method ($\mathcal{A_F V_R}$) contains lip-unsynced deepfake videos.

The aforementioned face-swapping methods are the most popular and most recent deepfake generation methods. In particular, Wav2Lip was chosen because of its efficiency, lesser time consumption, and better quality output (see Figure 3). The sample results can be seen in the bottom-left block in Figure 1. We generated 9,000 deepfake videos of this type. Attackers can employ this type of deepfake for identity fraud, making a person say anything by reenacting the face given any input audio, forging a person's image by swapping his or her face with someone else. Since we kept the audio intact for this type, i.e., used real audio, and manipulation is conducted with the video only, the audio is perfectly lip-synced with video. Researchers can use this $\mathcal{A_R V_F}$ dataset to train their detection models for a possible forged or face-swapped video detection.

**Fake-Audio and Fake-Video ($\mathcal{A_F V_F}$).** This type of dataset contains both fake video and respective fake audio. It combines the two types mentioned above ($\mathcal{A_F V_R}$ and $\mathcal{A_R V_F}$). See the bottom-right block in Figure 1 for sample images. We employ four types of deepfake generation methods, FaceSwap and FSGAN for face-swapping, Wav2Lip for audio-driven facial reenactment, and SV2TTS for real-time cloning a person's voice (see $\mathcal{A_F V_F}$ in Table 2). We first generate cloned fake audio using the SV2TTS tool by giving a text-audio pair as input. Then, we apply Wav2Lip to reenact the video based on cloned audio (see Figure 1). This type contains 10,000 deepfake videos. To the best of our knowledge, this type of deepfake dataset has not been released in the past.

## 3.2 Overall Dataset Generation Pipeline

A pictorial representation of our FakeAVCeleb generation pipeline is provided in Figure 3. For $\mathcal{A_F V_R}$, we begin with a source and target inputs having real audio and video. Then, we extract text from audio using IBM Watson speech-to-text service and generate a cloned speech using SV2TTS by providing the target's speech signal and extracted audio as an input (see Figure 3 left block; Step 1–3). Lastly, we combine them to make the $\mathcal{A_F V_R}$, as shown in Step 5. The middle block in Figure 3 illustrates $\mathcal{A_R V_F}$, which is a typical deepfake video generation pipeline. The source and target videos are processed with a faceswap method such as FSGAN or reenactment method such as Wav2Lip [62] to create deepfakes, as shown in Step 1–4. We combine the previous two methods ($\mathcal{A_F V_R}$ and $\mathcal{A_R V_F}$) to create Audio-Video deepfakes in the $\mathcal{A_F V_F}$ block (right) in Figure 3. We use the videos from source and target for faceswaping. At the same time, we use the target's speech signal to generate a cloned speech using IBM Watson speech-to-text service and SV2TTS (Step 1–3). We combine this cloned speech with the face-swapped video or use Wav2Lip to enhance its quality further, as shown in Steps 4–5.

## 4    Benchmark Experiments and Results

In this section, we present our experimental setup along with the detailed training procedures. We report the performance of different state-of-the-art baseline deepfake detection methods and discuss their limitations. The purpose of this comprehensive benchmark performance evaluation is to show

the complexity levels and usefulness of our FakeAVCeleb, compared to various other deepfake datasets.

**Preprocessing.** To train our baseline evaluation models, we first preprocess the dataset. Preprocessing was performed separately for videos and audios. Since we collected videos from VoxCeleb2 dataset, these videos are already face-centered. For audios, we compute MFCC features per audio frame and store them as a three channel image, which is then passed to the model as an input. The details of preprocessing are provided in Section B in Appendix.

**DeepFake Detection Methods.** To perform our experiments, we employed eight different deepfake detection methods to compare ours with other deepfake datasets, which are Capsule [13], HeadPose [14], Visual Artifacts (VA)-MLP/LogReg [51], Xception [36], Meso4 [15], and MesoInception [15]. We chose these methods based on their code availability, and performed detailed experiments to compare the detection performance of our FakeAVCeleb with other deepfake datasets. We have briefly explained each of seven baseline deepfake detection methods in Section A in Appendix. Furthermore, since our FakeAVCeleb is a multimodal dataset, we evaluated our model with two different techniques, i.e., the ensemble of two networks and multimodal-based detection, and used audios along with the frames as well. For ensemble networks, we used Meso4, MesoInception, Xception, as well as the latest SOTA models such as Face X-ray [65], F3Net [66] and LipForensics [67]. The details of these models and the detection performances are provided in Appendix. In particular, the detection performance of Face X-ray, F3Net, and LipForensics are 53.5%, 59.8%, and 50.4%, respectively. Therefore, the latest SOTA detection models have achieved mediocre or low detection performance, in contrast to their high detection performance on existing deepfake datasets such as UADFV, DF-TIMIT, FF++, DFD and DFDC.

**Results and Analysis.** The results of this work are based on the FakeAVCeleb v1.2 database. In the future, we will release new versions of FakeAVCeleb as the dataset is updated and further improved. Please visit our GitHub link[2] for the most recent results for each baseline on newer versions of FakeAVCeleb. The results for each experiment are shown in Table 3. We evaluate the performance of each detection method using the Area Under the ROC curve (AUC) at the frame-level. We use the frame-level AUC scores, because all the compared methods utilize individual frames and output a classification score. Moreover, all the compared datasets contain video frames and not respective audios. Table 3 shows AUC scores for each detection model over eight different deepfake dataset, including ours. Lower AUC score represents higher complexity of the dataset to be detected by the model. It can be observed that our FakeAVCeleb have the lowest AUC values mostly, and is closer to the performance of Celeb-DF dataset.

The classification scores for each method are rounded to three digits after the decimal point, i.e., with a precision of $10^{-3}$. Figure 4 presents the frame-level ROC curves of Xception, MesoInception4, and Meso4 on several datasets. The ROC curves in Figure 4 are plotted using only video frames for all dataset except our FakeAVCeleb, in which we used both audio (MFCCs) and video (frames) with our ensemble-based models of Xception, MesoInception4, and Meso4. The average AUC performance of all detection methods on each dataset is provided in Figure 6 in Appendix. It can be observed that our FakeAVCeleb is generally the most complex and challenging to these baseline detection methods, compared to all other datasets. The overall performance is close to that of Celeb-DF dataset. The detection performance of DFDC, DFD, and Celeb-DF methods is pretty low on recent datasets, producing average AUC less than 70%. Some of the detection methods achieve very high AUC scores on early datasets (UADF, DF-TIMIT, and FF-DF) with average AUC score more than 75%. However, the average AUC of our FakeAVCeleb is around 65%, where these averages are based on frame-level AUC scores and do not consider the audio modality.

For the performance of individual detection method, Figure 5 in Appendix shows the average AUC performance of each detection method on all evaluated datasets. Moreover, since the videos downloaded from online platform undergo compression because of the upload and distribution that causes change in video quality, we also show frame-level AUC of Xception-comp on medium (15) degrees of H.264 compressed videos of our FakeAVCeleb in Table 3. The results show that the detection performance of the compressed version of the dataset is close to raw dataset. Interested readers may refer to [68] for a more in-depth examination of FakeAVCeleb in unimodal, multimodal, and ensemble settings.

---

[2]https://github.com/DASH-Lab/FakeAVCeleb

Table 3: Frame-level AUC scores (%) of various methods on compared datasets. Bold faces correspond to the top performance.

| Dataset | UADFV | DF-TIMIT LQ | DF-TIMIT HQ | FF-DF | DFD | DFDC | Celeb-DF | FakeAVCeleb |
|---|---|---|---|---|---|---|---|---|
| Capsule [13] | 61.3 | 78.4 | 74.4 | 96.6 | 64.0 | 53.3 | 57.5 | 70.9 |
| HeadPose [14] | 89.0 | 55.1 | 53.2 | 47.3 | 56.1 | 55.9 | 54.6 | 49.0 |
| VA-MLP [51] | 70.2 | 61.4 | 62.1 | 66.4 | 69.1 | 61.9 | 55.0 | 67.0 |
| VA-LogReg [51] | 54.0 | 77.0 | 77.3 | 78.0 | 77.2 | 66.2 | 55.1 | 67.9 |
| Xception-raw [36] | 80.4 | 56.7 | 54.0 | 99.7 | 53.9 | 49.9 | 48.2 | 71.5 |
| Xception-comp [36] | **91.2** | **95.9** | **94.4** | **99.7** | **85.9** | 72.2 | **65.3** | **72.5** |
| Meso4 [15] | 84.3 | 87.8 | 68.4 | 84.7 | 76.0 | **75.3** | 54.8 | 60.9 |
| MesoInception4 [15] | 82.1 | 80.4 | 62.7 | 83.0 | 75.9 | 73.2 | 53.6 | 61.7 |

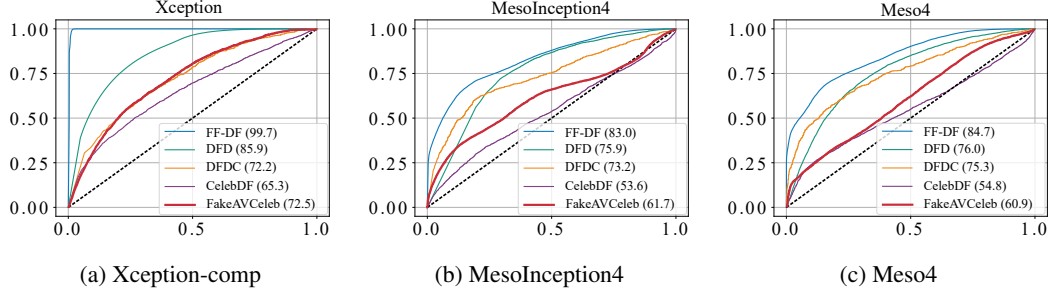

(a) Xception-comp  (b) MesoInception4  (c) Meso4

Figure 4: ROC curves of three state-of-the-art detection methods on five deepfake datasets. Only video frames are used from all datasets except FakeAVCeleb, where we used both audio (MFCCs) and video (frames) to form an ensemble model (see Appendix for more results). The AUC scores of these SOTA models on our FakeAVCeleb are 72.5%, 61.7%, and 60.9%.

## 5 Discussion and Future Work

**Data Quality.** In the process of collecting and generating our FakeAVCeleb dataset, we tried to ensure and maintain the high quality of our dataset. All of the selected real YouTube videos went through a manual screening process in which we carefully selected those videos having high quality and faces center-aligned and not covered. The generated deepfake videos also went through the same rigorous process in which we manually removed the corrupted videos. Before we apply face-swapping methods, we employed a facial recognition service, Face++ [63], in order to measure the similarity between two faces, and then applied face-swapping methods between faces having the highest similarity score.

**Data Availability and Social Impact.** Since we used the deepfake video and audio generation methods that are open-sourced and anyone can access and use them, we are not releasing a separate code repository. Please visit the DOI link [3], which contains the information related to FakeAVCeleb such as structure and version history of our dataset. While our dataset is openly available, it can be misused to evade existing deepfake detectors. To stop or limit the misuse of our FakeAVCeleb by bad actors, we have made a dataset request form[4]. We review the requests that we receive and allow access for a legitimate use. The dataset we share contains the real and all types of deepfake videos that we generated as a zip file. The package also contains the detailed documentation with all relevant metadata specified to users.

**Future Directions.** We plan to provide future updates to FakeAVCeleb, keeping updated with the latest deepfake video and audio generation methods. Also, we will consider the potential adversarial attacks and construct the dataset accordingly. Another area of improvement is that we will make use of recent deepfake polishing methods that will help remove the certain artifacts caused by deepfake generation methods. Since this is the first version of the dataset of this type and covers a variety of video and audio combinations with multiple deepfake generation methods, the number of deepfake videos may be small in numbers as compared to other large-scale deepfake datasets. We plan to increase the dataset size in future releases by utilizing more generally and publicly accessible videos

---

[3]http://doi.org/10.23056/FAKEAVCELEB_DASHLAB
[4]https://bit.ly/38prlVO

such as YouTube-8M [69] in our future release. And we will include them in our dataset maintenance plan.

## 6 Conclusion

We present a novel Audio-Video multimodal deepfake dataset, FakeAVCeleb, which can be used to detect not only deepfake videos but deepfake audios as well. Our FakeAVCeleb contains deepfake videos along with the respective synthesized cloned audios. We designed FakeAVCeleb to be gender and racially unbiased as it contains videos of men and women of four major races across different age groups. We employed a range of recent, most popular deepfake video and audio generation methods to generate nearly perfectly lip-synced deepfake videos with respective fake audios. We compare the performance of our FakeAVCeleb dataset with seven existing deepfake detection datasets. We performed extensive experiments using several state-of-the-art methods in unimodal, ensemble-based, and multimodal settings (see Appendix C for results). We hope FakeAVCeleb will help researchers build stronger deepfake detectors, and provide a firm foundation for building multimodal deepfake detectors.

## Broader Impact

To build a strong deepfake detector, a high-quality and realistic deepfake dataset is required. Recent deepfake datasets contain only forged videos or synthesized audio, resulting in methods for detecting deepfakes that are unimodal. Our FakeAVCeleb dataset includes deepfake videos as well as synthesized fake audio that is lip-synced to the video. To generate our FakeAVCeleb, we used four popular deepfake generation and synthetic voice generation methods. We believe that the multimodal deepfake dataset we are providing will aid researchers and open new avenues for multimodal deepfake detection methods.

## Acknowledgments and Disclosure of Funding

We thank Jeonghyeon Kim, Taejune Kim, Donggeun Ko, and Jinyong Park at Sungkyunkwan University for helping the dataset generation and validation, which greatly improves the current work. This work was partly supported by Institute of Information & communications Technology Planning & Evaluation (IITP) grant funded by the Korea government (MSIT) (No.2019-0-00421, AI Graduate School Support Program (Sungkyunkwan University)), (No. 2019-0-01343, Regional strategic industry convergence security core talent training business) and the Basic Science Research Program through National Research Foundation of Korea (NRF) grant funded by Korea government MSIT (No. 2020R1C1C1006004). Also, this research was partly supported by IITP grant funded by the Korea government MSIT (No. 2021-0-00017, Core Technology Development of Artificial Intelligence Industry), and was partly supported by the MSIT (Ministry of Science, ICT), Korea, under the High-Potential Individuals Global Training Program (2020-0-01550) supervised by the IITP (Institute for Information & Communications Technology Planning & Evaluation).

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
