# OpenReview forum: "FakeAVCeleb: A Novel Audio-Video Multimodal Deepfake Dataset"
_NeurIPS.cc/2021/Track/Datasets_and_Benchmarks/Round2 — NeurIPS 2021 Datasets and Benchmarks Track (Round 2)_

### Official Review · Reviewer_Z6Yw · 2021-09-11
**A New Audio-Video Multimodal Dataset With Diverse Combinations of Manipulations for Deepfake Detection**

**Rating:** 6
**Confidence:** 5

**Strengths:**

- The potential misuse and harm of advanced deep learning-based manipulation approaches (*i.e.*, Deepfakes) is a pretty significant issue. The introduced dataset can serve as a good contribution to alleviate this problem. A multimodal dataset including fake audio manipulation is currently missing. This work tries to fill this gap, which is useful for academia and industry.

- The paper is generally clearly written, and the dataset construction information is provided in detail.

- Representative related studies have been appropriately cited and discussed.

- The authors provide benchmark results on various datasets, including the proposed FakeAVCeleb, using seven baseline methods. The results demonstrate the complexity and challenge of the proposed dataset.

**Weaknesses:**

- The benchmark verified the complexity and challenge of the proposed dataset. However, it would be better if the authors could actually examine the usefulness of FakeAVCeleb, e.g., providing statistics to show the improved performance of forgery detectors with the help of the additional multimodal information. The authors can make the background introduction more concise for additional space.

- Reporting the exact number of fake videos and total videos in FakeAVCeleb is important, and the number of each type in Figure 1 should also be stated clearly.

- Please add the citation for each dataset in Table 1. Besides, Google DFD also contains real videos of source actors [1], and the "Rights Cleared" of DeeperForensics should be Yes. In addition, it is better to add one column about whether containing fine-grained labeling of fake videos and audios to show the superiority of FakeAVCeleb over DFDC.

- In the benchmark, the baselines are a bit old. It would be better if the authors could provide results on more state-of-the-art baselines, such as Face X-ray [2], F3-Net [3], LipForensics [4], *etc*.

---
[1] https://github.com/ondyari/FaceForensics/tree/master/dataset#space-requirement

[2] Face X-ray for More General Face Forgery Detection.  In CVPR 2020.

[3] Thinking in Frequency: Face Forgery Detection by Mining Frequency-aware Clues. In ECCV 2020.

[4] Lips Don't Lie: A Generalisable and Robust Approach to Face Forgery Detection. In CVPR 2021.

**Additional Feedback:**

- The current version of FakeAVCeleb only contains videos on celebrities. Indeed, celebrities are the most common objects in Deepfakes. However, it would be better if the dataset could include more diverse objects (i.e., not only celebrities but ordinary people) in the future. Also, the languages of speaking and the fake audio generation methods can be further expanded.

- Why are existing Deepfake datasets racially biased? For example, DeeperForensics contains actors from 26 countries, covering four typical skin tones: white, black, yellow, brown.

- What about the lip-unsynced cases? Can they be recognized as fakes? The authors may discuss this.

- Line 95, 96 Typo: "UADF" -> "UADFV".

- Line 184, 303, 355 Typo: a missing space after "FakeAVCeleb".

- Line 322, 330 Typo: What are Figure 5 and Figure 6 in the main paper?

- Line 330 Typo: "indiidual" -> "individual".

**Clarity:**

The paper is generally clearly written and explained in detail. Some writing issues can be found in Weaknesses and Additional Feedback.

**Correctness:**

Most of the claims in the submission are correct. Please refer to Weaknesses and Additional Feedback for some of my concerns.

**Documentation:**

The authors have provided details on data collection, organization, and generation. Providing the exact amount of different types of videos in FakeAVCeleb is also necessary, as suggested in Weaknesses. Besides, some details of videos are missing, *e.g.*, resolution, duration, *etc*. Also, it would be beneficial to include more detailed documentation of dataset usage and maintenance plans.

**Ethics:**

From my perspective, there are few or no ethical concerns that warrant further discussion or review. As the authors kindly request suggestions on other means to limit the misuse of the proposed dataset, maybe releasing a strong forgery detector (current best) as a countermeasure for the fake data would help.

**Relation To Prior Work:**

The paper appropriately cited relevant studies and discussed how this work differs from previous contributions.

**Summary And Contributions:**

This paper presents a new audio-video multimodal dataset, FakeAVCeleb, for Deepfake detection. The dataset contains three different types of audio-video forgeries on the combinations of video or audio manipulations (*i.e.*, real video & fake audio, fake video & real audio, fake video & fake audio) with accurate lip synchronization using existing popular Deepfake generation methods. The authors try to carefully select the source videos to make them diverse in ethnic backgrounds. They further benchmark seven baseline methods on FakeAVCeleb and several existing datasets for Deepfake detection.

---

> ### Author Response · Authors · 2021-09-27
> **Response to Reviewer Z6Yw (Part 2)**
>
> **Q6. Racial Bias in Deepfake Dataset.**
>
> We thank the reviewer for the insightful concern. Not all existing deepfake datasets are racially biased; however, we find that datasets such as KoDF [8] and Deepfake TIMIT [10] claim to contain only a single ethnicity. The KoDF contains only Korean ethnicity. Similarly, Deepfake Timit [10] mentions that it contains only English-speaking Americans.
> In addition, some DF datasets, such as Celeb-DF [9] have an unbalanced number of videos for different ethnicities and gender classes. Moreover, Celeb-DF mainly contains western celebrities. Therefore, we think it is partially biased. And [12] and [13] briefly discuss racial bias in the deepfake dataset, where [13] mentions that Celeb-DF contains mainly western celebrities. Also, we acknowledge that DeeperForensics [11] is not racially biased as it consists of actors from 26 countries. We have clarified this and toned down the sentences in the revised version on page 4. Additionally, it is important to mention that the number of videos belonging to each gender and ethnicity in existing deepfake datasets is different, which can result in a bias toward a specific race or gender when a classifier is trained on them whereas, in our FakeAVCeleb dataset we kept these number to be the same.
>
> **Q7. Lip-unsynced Cases.**
>
> We are grateful to the reviewer, raising the interesting and important question. Yes, the lip-unsynced cases can also be recognized as fakes. The multimodal detection methods should consider all modalities, and if any one of these modalities is fake, we consider the video as fake. We clarified this in the revised version. Furthermore, even though the lip-unsynced cases may look unnatural, we have considered them for an experiment along with other types and show the results in Appendix Figure 8. In this experiment, we train multiple models, one for each modality, then ensemble them. We find that fake audio in lip-unsynced cases is the primary factor in detecting this type of deepfakes.
>
>
> **Q8. Documentation.**
>
> We thank the reviewer for the comment on the documentation. We have included the missing details such as resolution and duration on page 4. Moreover, we will include more details on our long-term plan for the maintenance and expansion of FakeAVCeleb.
>
> **Q9. Figure 5 and Figure 6 location.**
>
> We apologize for the confusion caused to the reviewer. Figures 5 and 6 are shown in Appendix. We have fixed the text in the revised version.
>
> **Q10. Typos & Mistakes.**
>
> We really appreciate the reviewer for pointing out these typos after careful reviews. We have fixed all the typos and errors in the revised version.
>
>
> **References**
>
> - **[1]** Multimodal classification, 2019. URL:https://github.com/xkaple01/multimodal-classification
>
> - **[2]** Multimodal for movie genre prediction, 2021. URL:https://github.com/dh1105/Multi-modal-movie-genre-prediction.
>
> - **[3]** Yu, Zitong, et al. "Searching central difference convolutional networks for face anti-spoofing." Proceedings of the IEEE/CVF Conference on Computer Vision and Pattern Recognition. 2020.
>
> - **[4]** Li, Lingzhi, et al. "Face x-ray for more general face forgery detection." Proceedings of the IEEE/CVF Conference on Computer Vision and Pattern Recognition. 2020.
>
> - **[5]** Qian, Yuyang, et al. "Thinking in frequency: Face forgery detection by mining frequency-aware clues." European Conference on Computer Vision. Springer, Cham, 2020.
>
> - **[6]** Haliassos, Alexandros, et al. "Lips Don't Lie: A Generalisable and Robust Approach To Face Forgery Detection." Proceedings of the IEEE/CVF Conference on Computer Vision and Pattern Recognition. 2021.
>
> - **[7]** Abu-El-Haija, Sami, et al. "Youtube-8m: A large-scale video classification benchmark." arXiv preprint arXiv:1609.08675 (2016).
>
> - **[8]** Kwon, Patrick, et al. "KoDF: A Large-scale Korean DeepFake Detection Dataset." arXiv preprint arXiv:2103.10094 (2021).
>
> - **[9]** Li, Yuezun, et al. "Celeb-df: A large-scale challenging dataset for deepfake forensics." Proceedings of the IEEE/CVF Conference on Computer Vision and Pattern Recognition. 2020.
>
> - **[10]** Korshunov, Pavel, and Sébastien Marcel. "Deepfakes: a new threat to face recognition? assessment and detection." arXiv preprint arXiv:1812.08685 (2018).
>
> - **[11]** Jiang, Liming, et al. "Deeperforensics-1.0: A large-scale dataset for real-world face forgery detection." Proceedings of the IEEE/CVF Conference on Computer Vision and Pattern Recognition. 2020.
>
> - **[12]** Pu, Jiameng, et al. "Deepfake Videos in the Wild: Analysis and Detection." Proceedings of the Web Conference 2021. 2021.
>
> - **[13]** Tariq, Shahroz, Sowon Jeon, and Simon S. Woo. "Am I a Real or Fake Celebrity? Measuring Commercial Face Recognition Web APIs under Deepfake Impersonation Attack." arXiv preprint arXiv:2103.00847 (2021).

---

> > ### Comment · Reviewer_Z6Yw · 2021-10-03
> > **Most Concerns are Addressed. Additional Question about Lip-Unsynced Cases**
> >
> > Thanks for the detailed response. The rebuttal addresses most of my concerns.
> >
> > One additional question about the lip-unsynced cases: Could the authors provide some insights about the lip-unsynced cases where both the video and audio are real?

---

> > > ### Author Response · Authors · 2021-10-03
> > > **Response to Reviewer Z6Yw (Additional Question about Lip-Unsynced Cases)**
> > >
> > > **Question about Lip-Unsynced Cases**
> > >
> > > We thank the reviewer for raising this additional concern.
> > >
> > > We have not considered the scenario when both audio and video are real but unsynced (i.e., real video from person A and real audio from person B). It is an interesting scenario. However, since both the audio and video are real, we cannot put them in any of the fake categories (i.e.,  $\mathcal{A_FV_R}$,  $\mathcal{A_RV_F}$, and  $\mathcal{A_FV_F}$).
> > >
> > > Moreover, the lip-unsynced cases occur when we generate deepfakes, and we do not apply Wav2Lip. For example,  $\mathcal{A_FV_R}$ can contain unsynced lip cases.
> > >
> > > In the case of  $\mathcal{A_RV_R}$, they are lip-synced since the videos and audios are real, which are selected from the VoxCeleb-2 dataset.
> > >
> > > Please let us know if we understood your question correctly and if you want us to provide clarification.
> > >
> > >
> > >
> > > **These are the full form of the abbreviation:**
> > >
> > >  $\mathcal{A_RV_R}$: Real Audio and Real Video
> > >
> > >  $\mathcal{A_FV_R}$: Fake Audio and Real Video
> > >
> > >  $\mathcal{A_RV_F}$: Real Audio and Fake Video
> > >
> > >  $\mathcal{A_FV_F}$: Fake Audio and Fake Video

---

> > > > ### Comment · Reviewer_Z6Yw · 2021-10-04
> > > > **Additional Response and Reviewer Conclusion**
> > > >
> > > > The paper provided sufficient details clearly, and certainly, the reviewer did not miss these details. The lip-unsynced cases with real video and audio are not an additional concern but an open question for discussion. The reviewer was just unsure whether these cases can be regarded as fakes and could be included in FakeAVCeleb in the future.
> > > >
> > > > Thanks for providing the further response. Considering the usefulness and potential impact of the proposed dataset for the Deepfake detection community, I am happy to accept this submission if other reviewers have no additional concerns.

---

> > > > > ### Author Response · Authors · 2021-10-04
> > > > > **Thank you for the constructive suggestion for the future work.**
> > > > >
> > > > > We are very grateful for the reviewer's suggestion for future work. Certainly, they are the interesting cases, which the reviewer suggested. We plan to clarify such cases in the final version, and plan to incorporate such lip-unsynced cases for our future research. We will consider constructing and releasing the additional dataset for our future work.
> > > > >
> > > > > Thank you again for the valuable feedback and insights for our discussion and future work!!!

---

> ### Author Response · Authors · 2021-09-27
> **Response to Reviewer Z6Yw (Part 1)**
>
> **Q1. Usefulness of FakeAVCeleb.**
>
> We thank the reviewer for the interesting suggestion. We also hoped that the multimodal detector could improve the detection performance as suggested. However, we learned existing multimodal detectors such as CDCN [1], Multimodal-1 [2], and Multimodal-2 [3] are not designed, and optimized to detect multimodal deepakes and showed the low performance in our experiments as shown in Figure 9 in Appendix. This result directly indicates that future research should focus on developing multimodal deepfake detectors.
> Moreover, to demonstrate the usefulness of the additional information and features provided by our multimodal dataset for deepfake detection, we performed an experiment using ensembles of unimodal methods. As shown in Figure 8 in Appendix, we find that ensembles of unimodal audio and unimodal video can achieve higher performance than models trained on a single modality (see Figure 7 in Appendix), which suggest that our dataset with multimodal aspects/features indeed helps with deepfake detection.
>
> **Q2. Exact Dataset Statistics.**
>
> We thank the reviewer for the insightful suggestion. We believe that it will improve the readability of our paper. We have edited the caption of Figure 1 and mentioned the exact number of videos in each category. We have also added the data statistics in Section 3.1 and Table 1, where we have provided detailed information about each dataset category.
>
> **Q3. Clarifying Table 1.**
>
> We thank the reviewer for the suggestion and for pointing out the mistakes. We have fixed Table 1 in the revised version and clearly mentioned the dataset names, and provided the references, respectively. We have also fixed the number of real videos of Google DFD and “Rights cleared” of DeeperForensics [11]. Moreover, we have added another column containing fine-grained labeling of fake videos and audios to show the superiority of our FakeAVCeleb over other deepfake datasets.
>
> **Q4. Results on New SOTA Detectors.**
>
> We thank the reviewer for this suggestion. We added those baselines to compare the complexity offered by our FakeAVCeleb with other deepfake datasets since these baselines were common among all other datasets. We also agree that the baselines are a bit old. Therefore, we have decided to perform experiments using more state-of-the-art baselines such as Face X-ray [4], F3-Net[5], and LipForensics[6]. The results are shown below:
>
> | Models   	| Face X-ray[4] 	| F3Net [5] 	| LipForensics[6] 	|
> |:-: | :-: | :-: | :-: 	|
> | Ensemble 	|      76.2     	|    76.9   	|       56.0      	|
>
> As shown, the detection performance of FaceXray [5], F3Net [6], and LipForensics [7] are 76.2%, 76.9%, and 56%, respectively. Therefore, the latest SOTA detection models have difficulty achieving high detection performance, in contrast to their high detection performance on existing DF datasets.
> We also include these results in the revised version of the paper in Figure 8 in the Appendix.
>
> **Q5. More Diverse Subjects in Dataset.**
>
> We really appreciate the suggestion and we plan to generate deepfakes using the videos of the general public from datasets such as Youtube-8M [7] by Google research in our future release. And we will include them in our dataset maintenance plan.

---

### Official Review · Reviewer_kdNK · 2021-09-17
**A Novel Audio-Video Multimodal Deepfake Dataset**

**Rating:** 6
**Confidence:** 2
**Correctness:** Mostly correct.
**Clarity:** Yes. The paper is well written.

**Strengths:**

1) The proposed Audio-Video Deepfake dataset (FakeAVCeleb) contains not only deepfake videos but also respective synthesized lip-synced fake audios, which is an interesting and vital task for security and privacy issues.
2)

**Weaknesses:**

The reviewer was never involved in the related Deepfake field, thus only provides coarse insights.
1) Contribution/Comparison to prior datasets. The core of the work is the multimodal deepfake dataset with audio and video, but not much new insight in the experiments to support the contribution. Just using the existing DeepFake Detection Methods to obtain results on the proposed benchmark.
2) The author provides coarse data statistics(e.g., 490+, 20,000) in Table 1. The detailed information for the dataset is important for a dataset paper.

**Additional Feedback:**

None.

**Documentation:**

Yes.

**Ethics:**

Yes.

**Relation To Prior Work:**

Yes. The paper clearly discusses the relation to prior works.

**Summary And Contributions:**

- The authors present a novel Audio-Video Multimodal Deepfake Detection dataset, which contains both, video and audio deepfakes with accurate lip-sync, and three different types of Audio-Video deepfakes
- The authors provide a comprehensive baseline benchmark evaluation and demonstrated the crucial need and usefulness for a multimodal deepfake dataset.

---

> ### Author Response · Authors · 2021-09-27
> **Response to Reviewer kdNK (Part 2)**
>
> **Q2. Coarse Data Statistics.**
> We thank the reviewer for requesting this clarification. We have added the exact numbers and data statistics in Section 3.1 and Table 1. Also, we have provided detailed information about each dataset category.
>
> **References:**
>
> - **[1]**  Nguyen, Huy H., Junichi Yamagishi, and Isao Echizen. "Use of a capsule network to detect fake images and videos." arXiv preprint arXiv:1910.12467 (2019).
>
> - **[2]** Yang, Xin, Yuezun Li, and Siwei Lyu. "Exposing deep fakes using inconsistent head poses." ICASSP 2019-2019 IEEE International Conference on Acoustics, Speech and Signal Processing (ICASSP). IEEE, 2019.
>
> - **[3]** Rossler, Andreas, et al. "Faceforensics++: Learning to detect manipulated facial images." Proceedings of the IEEE/CVF International Conference on Computer Vision. 2019.
>
> - **[4]** Afchar, Darius, et al. "Mesonet: a compact facial video forgery detection network." 2018 IEEE International Workshop on Information Forensics and Security (WIFS). IEEE, 2018.
>
> - **[5]** Li, Lingzhi, et al. "Face x-ray for more general face forgery detection." Proceedings of the IEEE/CVF Conference on Computer Vision and Pattern Recognition. 2020.
>
> - **[6]** Qian, Yuyang, et al. "Thinking in frequency: Face forgery detection by mining frequency-aware clues." European Conference on Computer Vision. Springer, Cham, 2020.
>
> - **[7]** Haliassos, Alexandros, et al. "Lips Don't Lie: A Generalisable and Robust Approach To Face Forgery Detection." Proceedings of the IEEE/CVF Conference on Computer Vision and Pattern Recognition. 2021.

---

> ### Author Response · Authors · 2021-09-27
> **Response to Reviewer kdNK (Part 1)**
>
> **Q1. Existing Deepfake Detection Methods.**
>
> We thank the reviewer for bringing up the careful concern. Our main objective for this work is to provide a standardized benchmark dataset that can be evaluated against different deepfake detection algorithms. And there is less emphasis on developing a novel deepfake detection algorithm, which is not the primary focus of our paper. Therefore, for benchmarking comparison, we used state-of-the-art (SOTA) Deepfake detection methods such as Capsule [1], HeadPose [2], Xception [3], MesoInception4 [4] and others. Also, to compare our work with recent deepfake datasets such as CelebDF, we used the same baselines as used by these deepfake datasets to have a fair comparison.
>
> Moreover, we also experimented with the following recent SOTA deepfake detectors: FaceXray [5], F3Net[6], and LipForensics [7] as requested by the reviewer, and obtained the ensemble result for each method. For F3Net [6], the ensemble network is made using two F3Net models, one for each modality, i.e., audio and video. Meanwhile, for FaceXray and LipForensics, an ensemble network was made using original models with Xception [3] as the backbone classifier. The ensemble results from FaceXray [5], F3Net [6], and LipForensics [7] are 76.2%, 76.9%, and 56%, respectively. Therefore, the latest SOTA detection models have difficulty achieving high detection performance, as they did for other existing DF datasets.
>
> | Models   	| Face X-ray[5] 	| F3Net [6] 	| LipForensics[7] 	|
> |:-: | :-: | :-: | :-: 	|
> | Ensemble 	|      76.2     	|    76.9   	|       56.0      	|

---

### Official Review · Reviewer_di1u · 2021-09-21
**Better dataset for deepfake detection**

**Rating:** 7
**Confidence:** 4

**Strengths:**

- Novelty in the dataset as it is a AV multimodal dataset
- Using publicly available and widely used datasets like VoxCeleb2 dataset for their data generation process
- Using latest  and sota methods for deepfake data generation

**Weaknesses:**

Using publicly available datasets and models comes with the downside of acquiring the limitations of those datasets into their own dataset such as bias, which they tried to address by making the dataset as diverse as they could.

**Additional Feedback:**

If you could, in near future, provide the data generation pipeline for public use that will be a great contribution.

**Clarity:**

This paper has clearly describe the intend of the authors and what they were trying to achieve along with proper documentation and code for the general public understanding.

**Correctness:**

Yes the claims made in the paper seems to correct except the calling of existing deepfake datasets racially biased for which they could've provided a study supporting it or named the specific datasets which a biased.

**Documentation:**

The authors have provided all the sufficient information to obtain and use the dataset along with the code used to achieve the results.

**Ethics:**

As per my understanding this dataset does not raise any ethical concerns.

**Relation To Prior Work:**

This paper clearly furthers the previous efforts in this domain by creating a novel multimodal dataset.

**Summary And Contributions:**

This paper presented a novel AV multimodal deepfake detection dataset, which will further the deepfake detection. This dataset has both audio and video with audio lip-synced with the video, unlike other multimodal dataset available. This paper also focused on removing bias unlike other publicly available dataset by choosing people from different ethnic backgrounds. The results from this paper indicate the need of similar multimodal datasets to further this area of research. Although they've released the dataset if they could've provided the data generator itself it would have been helpful for the people indulging in this area of research.

---

> ### Author Response · Authors · 2021-09-27
> **Response to Reviewer di1u**
>
>
> **Q1. Bias in Public Datasets.**
>
> We thank the reviewer for the careful question. Yes, we acknowledge that these large-scale datasets (VoxCeleb) might possibly have gender/racial bias. We tried our best to remain unbiased by manually selecting an equal number of videos of people belonging to five different ethnicities to construct a fair dataset. On the other hand, the VoxCeleb is a huge dataset containing more than a million videos. We can manually add more samples, and increase the dataset size for each ethnicity and gender (male & female), while remaining unbiased.
>
>
> **Q2. Correctness 一 Racially Biased Datasets.**
>
> We thank the reviewer for the insightful question and suggestions. The deepfake datasets that are known to be partially biased include UADFV [1], Deepfake TIMIT [2], and KoDF [3]. The KoDF contains only Korean ethnicity. Similarly, UADFV mainly contains 49 real videos from Youtube, and Deepfake Timit mentions that it contains only English-speaking Americans.  Moreover, some DF datasets such as Celeb-DF [4] have an unbalanced number of videos for different ethnicity and gender classes. To the best of our knowledge, no extensive study explores the racial bias in the deepfake dataset. However, [5] and [6] briefly discuss racial bias in deepfake dataset. Moreover, [5] mentions that Celeb-DF contains mainly western celebrities. Based on the reviewer's suggestions, we have fixed the text in the revised version and now clearly mention which deepfake datasets are biased (page 4).
>
>
> **Q3. Dataset Generation Pipeline.**
>
> We thank the reviewer for this suggestion. We are happy to provide the dataset generation pipeline on Github for public use. However, we are concerned that if we release the entire pipeline publicly, it might be used by the bad actors for malicious purposes. Currently, we only provide benchmarking codes. One alternative is to provide the pipeline upon request. We will review the requests that we receive and allow access for only legitimate use (e.g., emails from educational institutes or PI). We appreciate any suggestions from the reviewers on this matter.
>
> **References:**
>
> - [1] Yang, Xin, Yuezun Li, and Siwei Lyu. "Exposing deep fakes using inconsistent head poses." ICASSP 2019-2019 IEEE International Conference on Acoustics, Speech and Signal Processing (ICASSP). IEEE, 2019.
>
> - [2] Korshunov, Pavel, and Sébastien Marcel. "Deepfakes: a new threat to face recognition? assessment and detection." arXiv preprint arXiv:1812.08685 (2018).
>
> - [3] Kwon, Patrick, et al. "KoDF: A Large-scale Korean DeepFake Detection Dataset." arXiv preprint arXiv:2103.10094 (2021).
>
> - [4] Li, Yuezun, et al. "Celeb-df: A large-scale challenging dataset for deepfake forensics." Proceedings of the IEEE/CVF Conference on Computer Vision and Pattern Recognition. 2020.
>
> - [5] Tariq, Shahroz, Sowon Jeon, and Simon S. Woo. "Am I a Real or Fake Celebrity? Measuring Commercial Face Recognition Web APIs under Deepfake Impersonation Attack." arXiv preprint arXiv:2103.00847 (2021).
>
> - [6] Pu, Jiameng, et al. "Deepfake Videos in the Wild: Analysis and Detection." Proceedings of the Web Conference 2021. 2021.

---

### Official Review · Reviewer_ygzx · 2021-09-23
**A dataset with clearly labeled fake videos and fake audios**

**Rating:** 6
**Confidence:** 4
**Correctness:** Yes.
**Clarity:** Yes, the paper is well written and ea…

**Strengths:**

1. The paper is well-written and organized. There is sufficient detail on data collection and processing procedures.
2. This is the first dataset with clearly labeled fake audio and videos. As malicious content can be created with just fake audio or video, the dataset can be useful to the forensics community.
3. The dataset is based on VoxCeleb, and one can expand the dataset more easily.
4. Face similarity matching using Face++, and manual inspection increases the quality of the dataset.
5. A comprehensive benchmarking is performed on the dataset. Comparisons against the prior DeepFake datasets are carried out, and the results suggest that FakeAVCeleb has a comparable complexity and difficulty against the recently published DeepFake datasets.

**Weaknesses:**

1. The released dataset only contains cropped faces. It would be good to also release a version of the dataset with the uncropped video. This can be helpful because in-the-wild fakes are uncropped, and different DeepFake detection algorithms can use different face detection methods. Some potential methods might not even use cropped faces as inputs; for example, image splicing detection.
2. For the fake audio, will it be better to also include identity swaps? Currently, my understanding is that fake audio is generated by an auto-encoding process, so the fake audio contains the text and sound profile from the same person. Would it be possible to include audio with text from person B and sound profile from person A? For the case of (real video, fake audio) pair, it should be more reasonable to use the same sound profile as the person in the real video.
3. The labels in Figure 3 will need better descriptions. For example, result (A) in the middle column seems to be before lip-synced. Is result (A) included in the dataset?

**Additional Feedback:**

Some typos:

line 303: FakeAVCelebwith -> FakeAVCeleb with

line 305: FakeAVCelebis -> FakeAVCeleb is

line 328: FakeAVCelebis -> FakeAVCeleb is

line 330: indiidual -> individual

line 334: FakeAVCelebin -> FakeAVCeleb in

line 350: FakeAVCelebby -> FakeAVCeleb by

line 355: FakeAVCelebstill -> FakeAVCeleb still

**Documentation:**

Yes, there is sufficient detail.

**Ethics:**

No, I am not aware of any ethical concerns.

**Relation To Prior Work:**

Yes.

**Summary And Contributions:**

This work introduces a novel DeepFake dataset containing both fake video and fake audio. Most prior DeepFake benchmarks do not contain fake audio; DFDC, one of the exceptions, contains fake audio with the video, but the dataset does not specify if the video is fake or the audio. The proposed dataset has collected a combination of $\left[ \text{real, fake video} \right] \times \left[ \text{real, fake audio} \right]$, and it is clearly specified whether the video or audio is fake. Fake audio is generated by an auto-encoding scheme with audio-text translation and TTS. Fake video is generated from a collection of common DeepFake methods. Prior datasets are compared against the proposed dataset using various detection methods, where the results suggest that FakeAVCeleb has a comparable complexity and difficulty against the recently published DeepFake datasets.

---

> ### Author Response · Authors · 2021-09-27
> **Response to Reviewer ygzx**
>
>
> **Q1. Uncropped Dataset.**
>
> We thank the reviewer for this very insightful suggestion. We agree that it would be helpful to release the uncropped videos, as suggested. Hence, we will release the dataset with uncropped videos with the same dimensions as the VoxCeleb dataset.  One can view  a sampled uncropped version of our dataset on the following Google drive link: https://drive.google.com/drive/folders/1SYMs44Z1W7rlrn0W7t-4LcPPusiBlNEB
>
>
> **Q2. Fake Audio for Identity Swap.**
>
> We thank the reviewer for the interesting question. In fact, we have generated the combination AFVR (fake audio:real video) with the settings mentioned by the reviewer, i.e., using the sound profile of person A (target) and audio with text from person B (source). However, we made a typo in Figure 3 (1st column) that we connect an arrow from person B (Green) instead of Person A (Blue). We apologize for the inconvenience and confusion caused to the reviewer, and we have fixed this in Figure 3, and updated it in the revised version. You can view the before and after on the following link:
> https://ibb.co/fSkLgFK
>
> **Q3. Figure 3 labeling and descriptions.**
>
> We thank the reviewer for bringing this to our attention. Yes, Result (A) is before the lip-sync. In order to consider diverse generation scenarios, we have included Result (A) in the dataset. To clarify and remove any future confusion for the reader, we have renamed the dataset folder and made it more consistent with the nomenclature used in the paper and Figure 3. For example, Result (A) of Real Audio & Fake Video (ARVF) in Figure 3 (Middle column) is present in this directory: RealAudio_FakeVideo(ArVf)/Results. You can view this updated dataset on the following Google drive link:
> https://drive.google.com/drive/folders/1SYMs44Z1W7rlrn0W7t-4LcPPusiBlNEB
>
> **Q4. Typos.**
>
> We really appreciate the reviewer for pointing out these typos. We have fixed all the typos and errors in the revised version.

---

> > ### Comment · Reviewer_ygzx · 2021-10-03
> > **Response to the Rebuttal**
> >
> > Thanks for the detailed response!
> >
> > After reading all the reviews and the authors' responses, all of my concerns are addressed, and I think that the author also addressed most of the other reviewers' comments. I would recommend this paper to be accepted since this is a well-curated dataset and will be useful to the community.

---

> > > ### Author Response · Authors · 2021-10-03
> > > **Thanks to the Reviewer ygzx**
> > >
> > > We would like to express our sincere gratitude to the reviewer for the time and consideration of our work, as well as their insightful feedback. We are incredibly appreciative of their thoughtful comments and assessment of our work. We believe that this process significantly improved the work and manuscript's quality. Additionally, we would like to express our gratitude to the reviewer for the complimentary comments on the work's quality, the potential benefit to the field, and overall positive reception.

---

### Author Response · Authors · 2021-09-27
**Paper Revision: Summary of updates and thanks.**

We appreciate the reviewers for their constructive comments and valuable feedback. We have now revised the paper and made the requested changes and using the additional page. Also, we added more experimental results in Appendix. The changes are highlighted in yellow in the new version, and the summary of the changes are provided below:

- We have included the uncropped dataset samples and clarified the Fake Audio for Identity Swap case.

- We have clarified the lip-sync scenario in Figure 3 and provide the sample data.

- We have improved the presentation of our work and fixed all the typos and grammatical issues.

- We have provided more discussion and clarification on the bias and issue in the source dataset.

- In addition, we have conducted additional experiments using the three latest SOTA deepfake detection methods on our datasets and demonstrated the effectiveness of our dataset.

- We have provided the exact number of statistics on our dataset and information in Table 1.

- Also, we have discussed the lip-unsynched case and presented the result in Appendix.

Thanks again for your consideration and for giving us an opportunity to revise the paper for improvement. We will be happy to address any other concerns and questions reviewers may have and look forward to receiving your feedback and comments on this new version of our paper and our responses.

---

### Decision · Program_Chairs · 2021-10-09

**Decision:**

Accept

**Comment:**

The paper introduces a new dataset of DeepFake video, where each instance may have any combination of real or fake audio or video. Initial reviews were mixed, but the author responses and paper revisions were able to address all reviewer concerns and in the end all reviewers thought that the paper was above the bar for acceptance. Congratulations on having your paper accepted to the NeurIPS 2021 Datasets & Benchmarks Track!